# Decoy Receptor Interactions as Novel Drug Targets against EKC-Causing Human Adenovirus

**DOI:** 10.3390/v11030242

**Published:** 2019-03-12

**Authors:** Naresh Chandra, Lars Frängsmyr, Niklas Arnberg

**Affiliations:** Section of Virology, Department of Clinical Microbiology, Umeå University, SE-90185 Umeå, Sweden; naresh.chandra@umu.se (N.C.); lars.frangsmyr@umu.se (L.F.)

**Keywords:** glycosaminoglycans, adenovirus, cellular receptor, decoy receptor, epidemic keratoconjunctivitis, GAG-mimetic

## Abstract

Epidemic keratoconjunctivitis (EKC) is a severe ocular disease and can lead to visual impairment. Human adenovirus type-37 (HAdV-D37) is one of the major causative agents of EKC and uses sialic acid (SA)-containing glycans as cellular receptors. Currently, there are no approved antivirals available for the treatment of EKC. Recently, we have reported that sulfated glycosaminoglycans (GAGs) bind to HAdV-D37 via the fiber knob (FK) domain of the viral fiber protein and function as decoy receptors. Based on this finding, we speculated that GAG-mimetics may act as artificial decoy receptors and inhibit HAdV-D37 infection. Repurposing of approved drugs to identify new antivirals has drawn great attention in recent years. Here, we report the antiviral effect of suramin, a WHO-approved drug and a widely known GAG-mimetic, against HAdV-D37. Commercially available suramin analogs also show antiviral effects against HAdV-D37. We demonstrate that suramin exerts its antiviral activity by inhibiting the attachment of HAdV-D37 to cells. We also reveal that the antiviral effect of suramin is HAdV species-specific. Collectively, in this proof of concept study, we demonstrate for the first time that virus binding to a decoy receptor constitutes a novel and an unexplored target for antiviral drug development.

## 1. Introduction

Epidemic keratoconjunctivitis (EKC) is a severe ocular disease and is caused largely by species D human adenovirus type -8 (HAdV-D8), -D37, -D53, -D54, -D56, and -D64 [1,2]. HAdV infections are generally self-limiting but can also lead to serious life-threatening diseases in immunocompromised individuals [3]. HAdV-caused EKC may initially exhibit flu-like manifestations and displays severe ocular symptoms such as keratitis, conjunctivitis, eyelid edema, severe pain, excessive tearing, irritation, foreign body sensation, and the formation of pseudo-membranes [2]. The infection of the cornea by EKC-causing HAdVs activates the secretion of chemokines such as IL8 and MCP1 from infected corneal keratocytes [2]. These chemokines induce the infiltration of various immune cells (subepithelial infiltrates) into the corneal stroma, which is a hallmark of EKC. These infiltrates can persist from months to years and can lead to visual impairment [4,5]. EKC often begins as a unilateral condition (infection of one eye), but in most cases, it becomes bilateral as a result of eye-to-hand-to-eye transmission [2].

Although outbreaks of EKC occur globally on a regular basis, they are frequently reported in densely populated areas/countries [6,7,8]. Each year, around 20–30 million cases of HAdV-associated conjunctivitis are reported worldwide, with Japan alone having an incidence of more than one million cases [7,9]. Since individuals suffering from EKC are highly contagious, these patients are advised to refrain from the workplace, which causes big socio-economic loss, especially in developing countries [4]. No antiviral drugs are available for treatment of EKC or any other HAdV infection, thus antiviral therapies against HAdV infections are highly desirable. Nucleotide analogs, such as cidofovir and ganciclovir efficiently inhibit HAdV infection in vitro [10]. Unfortunately, side effects such as nephrotoxicity limit the scope of these inhibitors for clinical use [10]. Ribavirin also inhibits HAdV infection in vitro, but the antiviral activity of ribavirin is restricted to species C HAdVs [11]. Small trivalent sialic acids (SAs) are potent inhibitors of EKC-causing HAdV-D37 infection in vitro with low IC_50_ values (nM ranges) but are not yet available for clinical use [12,13]. Currently, most therapeutic interventions for EKC are primarily focused on limiting the severity of symptoms [2].

Blocking the attachment of viral pathogens to cellular receptors is an attractive approach for the development of antivirals since it impairs the very first step of the viral infection cycle. Many pathogenic viruses make use of cell-surface glycans such as SAs and glycosaminoglycans (GAGs) for initial attachment to target cells [14,15]. The binding of viruses to these glycans is often an outcome of charge-dependent interactions between negatively charged glycans and viral proteins that carry a positively charged surface/motif. Soluble, negatively charged glycan-based molecules can interfere with these interactions and inhibit virus attachment and subsequent infection of cells [16,17]. For example, heparin, a highly sulfated analog of heparan sulfate (HS), inhibits the infection of various GAG-binding viruses [18,19]. Most HAdVs use proteinaceous receptors and mediate interactions with cellular receptors via the knob domain of the viral capsid fiber protein [20,21,22]. However, EKC-causing species D HAdVs use SA-containing glycans for the attachment to and infection of cells [23]. EKC-causing HAdV-D37 binds (via its highly positively charged fiber knob (FK)) to SA-containing hexasaccharid, resembling those in GD1a-gangliosides, and uses these glycans as receptors for infection of ocular cells [24]. Interactions of HAdV-D37 with SA-glycans have been explored for the design and synthesis of multiple SA-based attachment inhibitors [12,13,25].

Recently, we reported a novel function of soluble and cell surface GAGs as decoy receptors for HAdV-D37 [26]. We showed that soluble, sulfated GAGs and cell surface-bound HS bind to incoming HAdV-D37 viruses and prevent/delay virus binding to SA-containing receptors on human corneal epithelial (HCE) cells that represent the ocular tropism of HAdV-D37. Based on this finding, we hypothesized that negatively charged GAG-mimetics can potentially function as “soluble super-decoys”, which can bind to the positively charged HAdV-D37 fiber knob (37FK) and disrupt the virus attachment to cells.

Drug repurposing is a promising approach for identifying new therapeutic agents against viral pathogens [27]. This approach offers several advantages such as the pharmacokinetic and toxicity profiles of approved drugs are well-documented. Altogether, this encouraged us to investigate the antiviral activity of a low molecular weight GAG-mimetic (i.e., suramin) against HAdV-D37. Suramin is an approved drug for the treatment of trypanosomiasis and onchocerciasis [28,29]. It also shows antiviral activity against a broad range of viruses with multiple modes of actions [30,31,32]. Suramin inhibits replication of retroviruses and other RNA viruses by inhibiting the activity of the reverse transcriptase and RNA polymerases, respectively [33,34]. It also blocks the attachment of herpes simplex virus-1, chicken guinea virus, and dengue virus in vitro [16,31,35].

Here, we report a novel anti-adenoviral activity of suramin and its analogs against HAdV-D37. We also show the mechanism of action of suramin at the viral attachment step. This new finding demonstrates that suramin, “a soluble super-decoy”, not only disrupts decoy receptor- (i.e., GAG-) dependent attachments but also functional receptor- (i.e., SA) dependent attachments of HAdV-D37 to cells.

## 2. Materials and Methods 

### 2.1. Cells, Viruses, Antibodies, Glycans, Suramin, and Suramin Analogs

Cells: Human corneal epithelial (HCE) cells (a gift from Dr. Araki-Sasaki) were cultured in SHEM medium (1:1; DMEM (Dulbecco’s Modified Eagle Medium) and HAM’s F12 Nutrient Mixture supplemented with 20 mM 4-(2-hydroxyethyl)-1-piperazine-ethane-sulfonic-acid (HEPES), 5 μg/mL insulin, 0.5% dimethyl sulfoxide (DMSO), 0.1 μg/mL cholera toxin, 10 ng/mL human Epidermal Growth Factor (Sigma), 20 U/mL penicillin + 20 μg/ml streptomycin (PEST, Invitrogen), and 10% fetal bovine serum (FBS). A549 cells (a gift from Dr. Alistair Kidd) were cultured in DMEM medium supplemented with 20 mM HEPES, 20 U/ml penicillin + 20 μg/mL streptomycin, and 10% FBS.

Viruses: HAdV-C5 (strain 75) and HAdV-D37 (strain 1477) viruses were propagated in A549 cells with and without ^35^S-labeling, as reported previously [17], except that viruses were eluted in phosphate buffer saline (PBS) on a NAP column (GE Healthcare, Chicago, IL, USA) and stored (at −20 °C) in PBS containing 10% glycerol.

Antibodies: Serotype-specific rabbit HAdV-C5 and HAdV-D37 antisera (gifts from Prof. Göran Wadell) were used for infection experiments. Mouse monoclonal anti-RGS-His (recognizing the epitope RGSHHHH; from Qiagen) was used for flow cytometry. Swine anti-rabbit IgG antibody fluorescein isothiocyanate- (FITC-) conjugate (from Dako Cytomation) and donkey anti-mouse IgG antibody Alexa Fluor 488 conjugate (from Invitrogen) were used as secondary antibodies for infection and flow cytometry experiments, respectively. Heparin (from porcine intestinal mucosa, product ID H3149), hyaluronic acid (HA; from *Streptococcus equi*, product ID 94137), suramin (product ID S2671), and NF023 (product ID N8652) were purchased from Sigma Aldrich. NF110 (cat. no. 2548) and NF449 (cat. no. 1391) were purchased from Tocris Biosciences. 

### 2.2. Cloning, Expression, and Purification of the Fiber Knob

Cloning, expression, and purification of fiber knobs (FKs) were carried out as described previously [36]. Briefly, for the production of His-tagged FKs, HAdV-C5 and HAdV-D37 FK genes were cloned into a pQE30-Xa expression vector encoding a His-tag (N-terminal) using restriction sites for BamHI and XmaI (Thermo Scientific, Waltham, MA, USA). For the production of GST-tagged 37FKs, HAdV-D37 FK gene was cloned into a pGEX-6P expression vector encoding a GST-tag (N-terminal) using restriction sites for NcoI and XhoI (Thermo Scientific, Waltham, MA, USA). All constructs were confirmed by sequencing (Eurofins MWG Operon, Ebersberg, Germany). His-tagged and GST-tagged proteins were expressed in *Escherichia coli* strain M15 and *E. coli* Rosetta strain (Qiagen, Helden, Germany), respectively. Proteins were expressed according to the protocol from Qiagen (The QIAexpressionist^TM^, Helden, Germany). Briefly, 3 L of bacterial culture were incubated at 37 °C to an optical density of 0.6. The culture was then induced with freshly prepared 1 mM isopropyl β-d-1-thiogalactopyranoside (IPTG; Thermo Scientific). After 4–5 h, the culture was pelleted and stored at −20 °C. His-tagged FKs were purified with Ni-NTA agarose beads, whereas GST-tagged FKs were purified with GST-sepharose beads followed by anion exchange (Q-sepharose) chromatography. The sizes and purities of proteins were examined by running the purified proteins on the SDS-PAGE gel (NuPAGE Bis-Tris; Invitrogen, Carlsbad, CA, USA).

### 2.3. Infection Assays

HCE cells were cultured as a monolayer in the transparent flat bottom (30,000 cells/well) 96-well plates. The cells were then washed twice with serum-free growth medium. Viruses were added to cells and incubated for 1 h on ice. To remove unbound viruses, cells were washed three times with serum-free growth medium. The cells were incubated for 44 h at 37 °C in culture medium containing 1% FBS. After 44 h incubation, the cells were washed once with PBS (pH 7.4) and fixed with ice-cold methanol. The cells were then incubated with polyclonal rabbit anti-HAdV-C5 (1:5000) and HAdV-D37 (1:150) antisera diluted in PBS for 1 h at room temperature (RT). The cells were washed three times with PBS and incubated for 1 h at RT with swine anti-rabbit IgG FITC-conjugated antibody (1:100) diluted in PBS. The cells were washed once with PBS and stained with DAPI (4′,6-diamidino-2-phenylindole; from Vector Laboratories, diluted 1:5000 in PBS) for 5 min. The cells were then washed twice with PBS. The infection of cells was analyzed in Trophos (Luminy Biotech Enterprises, Marseille, France). Infection assays were performed with the following variations: (i) HAdV-D37 viruses were treated with increasing concentrations of suramin and suramin analogs diluted in serum-free growth medium, (ii) HAdV-C5 and HAdV-D37 viruses were treated with suramin (1 mM) and heparin (0.4 mM) diluted in serum-free growth medium. Untreated viruses were used as control.

### 2.4. Virus Binding Assay

HCE cells were detached with pre-warmed PBS containing 0.05% ethylene-diamine-tetra-acetic acid (EDTA). The cells were counted and reactivated in 10% growth medium for 1 h at 37 °C (in suspension). The cells were then pelleted in a V-bottom 96-well plate (1 × 10^5^ cells/well) and washed once with binding buffer (BB; DMEM supplemented with 20 mM HEPES, 20 U/mL penicillin + 20 g/mL streptomycin and 1% bovine serum albumin). ^35^S-labeled HAdV-D37 viruses (10,000 vp/cell, diluted in BB) were added to cells and incubated for 1 h at 4 °C on ice. Viruses were pre-treated with increasing concentrations of suramin. Untreated viruses were used as control. To remove unbound viruses, cells were washed three times with BB. Cell-associated radioactivity was measured by using Wallac 1409 liquid scintillation counter (PerkinElmer, Waltham, MA, USA).

### 2.5. Cell Viability Assay

HCE cells (25,000 cells/well) were seeded in a 96-well plate. After 24 h, cell medium was removed and fresh medium containing increasing concentrations of suramin was added to cells. Cells grown without suramin were used as control. Cell viability was measured after 24 h by using the CellTiter-Glo Luminescent Cell Viability Assay Kit (cat. No. G7571; from Promega).

### 2.6. Fiber Knob Binding Assay

HCE cells were detached with pre-warmed PBS containing 0.05% EDTA. The cells were counted and then reactivated in 10% growth medium for 1 h at 37 °C (in suspension). After 1 h, cells (2 × 10^5^ cells/well) were pelleted in a V-bottom shaped 96-well plate and washed once with BB. The cells were then incubated with 10 μg/mL of 37FKs in 100 μL BB for 1 h at 4 °C on ice. FKs were pre-treated with heparin (0.4 mM) and suramin (1 mM). Untreated FKs were used as control. To remove unbound FKs, cells were washed twice with BB. The cells were then incubated with monoclonal mouse anti-RGS-His antibody (diluted 1:200 in BB) for 1 h at 4 °C on ice. After 1 h incubation, the cells were washed once with BB and incubated with donkey anti-mouse Alexa Fluor 488 antibody (diluted 1:1000 in BB) for 1 h at 4 °C on ice. Thereafter, the cells were washed with FACS buffer (PBS with 2% FBS) and analyzed with flow cytometry using a FACS LSRII instrument (Becton Dickinson, Franklin Lakes, NJ, USA). Results were analyzed using FACSDiva software (Becton Dickinson, Franklin Lakes, NJ, USA).

### 2.7. Surface Plasmon Resonance 

Surface plasmon resonance (SPR) was performed in a Biacore T200 instrument (GE, Chicago, IL, USA). Recombinant GST-tagged 37FKs were coupled to the CM5 sensor chip by using the amine coupling reaction according to the manufacturer’s instructions, resulting in an immobilization density of 7500–10,000 RU. The surface of the upstream flow cell was subjected to recombinant GST-coupling reaction protein and used as a reference. All binding assays were carried out at 25 °C. PBS-Tween was used as a running buffer. Analyte (suramin) was serially diluted in running buffer to prepare a two-fold concentration series ranging from 1000 to 390,625 µM and then injected in series over the reference (GST) and experimental biosensor surfaces (GST-tagged 37FK) for 60 s with a dissociation time of 60 s. Blank samples containing only running buffer were also injected under the same conditions to allow for double referencing. The binding affinity (KD) was calculated using BIAcore T200 evaluation software.

### 2.8. Statistical Analysis

All experiments were performed two or three times with duplicate or triplicate samples. All results are presented as standard error of the mean (SEM). Graphical and statistical analyses were performed by using GraphPad Prism version 7 for Windows (GraphPad Software, San Diego, CA, USA). Significance was calculated using Student ’s *t*-test. All *P*-values of <0.05 were considered statistically significant.

## 3. Results and Discussion

We have recently proposed that sulfated GAGs in secretions and on the cell surface trap HAdV-D37 and delay and/or prevent virus interaction with functional, SA-containing receptors [26]. We also suggested that sulfated GAG-mimetics can bind to/trap HAdV-D37 and can be explored as attachment inhibitors against HAdV-D37. To test this, we examined the effect of suramin, a known GAG-mimetic, on HAdV-D37 infection of ocular cells (i.e., HCE cells). Suramin is composed of eight linearly oriented benzene rings and contains six sulfate groups that are located on the terminal benzene rings (Figure 1A). To understand the effect of spatial arrangement and the number of benzene rings and the level of sulfation on the antiviral activity, we also included suramin analogs (NF023, NF0110, and NF449) in the infection assay (Figure 1B–D). NF023, NF0110, and NF449 contain six benzene rings each and are used as purinergic receptor antagonists [37,38]. Unlike suramin, these compounds are not approved drugs. NF023 displays similar orientation of sulfate groups as suramin, whereas sulfate groups on NF0110 and NF449 are equally distributed on terminal benzene rings.

Interestingly, pre-incubating HAdV-D37 with these compounds efficiently inhibited the virus infection of HCE cells in a dose-dependent manner (Figure 2A). Suramin inhibited HAdV-D37 infection most efficiently with an IC_50_ value of 65 µM, followed by NF023 (125 µM), NF449 (170 µM), and NF110 (240 μM). It is of interest that suramin inhibited HAdV-D37 infection with similar efficiency as reported for soluble GD1a-glycans (IC_50_: ~25 µM) [13], i.e., the cellular receptor for HAdV-D37. This result indicates that suramin and its analogs hold the ability to prevent HAdV-D37 infection of cells possibly by inhibiting virus attachment to cells. We hypothesize that the local concentration of sulfate groups around the two benzene rings present in suramin and the second most efficient inhibitor NF023 (but not in the other, less efficient compounds) contribute to the stronger interaction with the virus and their relatively higher inhibitory efficiencies. To elucidate if suramin, the most potent inhibitor, inhibits virus attachment to cells, we pre-incubated ^35^S-labeled HAdV-D37 with increasing concentrations of suramin and analyzed virion binding to HCE cells. Indeed, suramin inhibited virus binding to HCE cells in a dose-dependent manner (Figure 2B), confirming that suramin exerts its effect at the virus-attachment step. To exclude the possibility that suramin had reduced virus attachment and infection because of its cytotoxic effect, we performed cell viability assay. Cells treated for 24 h with increasing concentrations of suramin exerted no obvious cytotoxicity even at the highest (5 mM) concentration (Figure 2C).

Previous studies have suggested that suramin displays antiviral activity by disrupting critical charge-dependent interactions. For example, suramin binds to positively charged pocket of severe fever thrombocytopenia syndrome virus (SFTSV) nucleocapsid protein and occupies the RNA-binding cavity, which consequently hampers the virus replication [32]. Suramin also binds to the positively charged region of enterovirus-A71 capsid protein and disrupts virus binding to cell surface receptors (i.e., HS) [39]. Most HAdVs bind to primary attachment receptors via the knob domain of the trimeric fiber protein [40]. 37FK has an overall positive charge (isoelectric point; pI = 9.14; Table 1) and contains a highly positively charged SA-binding central cavity on the top of the FK [36]. This central cavity may favor either specific or unspecific, charge-dependent interactions with negatively charged sulfated molecules such as suramin. We speculated that suramin binds to the positively charged central cavity in the 37FK and probably occupies the SA-binding pocket. As a result, HAdV-D37 virions cannot access cell-surface SA-containing glycans, which consequently prevents the virus attachment to, and subsequent infection of, cells.

To elucidate this, we analyzed the effect of suramin on 37FK binding to HCE cells. We recently showed that heparin (a highly sulfated GAG) binds to 37FK and inhibits the knob binding to cells [26]; therefore, heparin was used as a control glycan in the assay. Pre-incubation of His-tagged 37FK with suramin or heparin inhibited 37FK binding to HCE cells (Figure 3A), indicating that suramin exerts (as heparin) its inhibitory effect by interfering with 37FK binding to cells. Further, to show the direct interaction and determine the affinity between suramin and 37FK, we performed surface plasmon resonance (SPR) analysis. We analyzed suramin (in solution) interaction with GST-tagged 37FKs (instead of His-tagged knobs, to avoid unspecific interactions) that were immobilized onto a CM5 sensor chip. Suramin displayed binding to 37FK with micromolar affinity (~204 μM) (Figure 3B), demonstrating the ability of suramin to directly bind to the viral FK and assumingly to the SA-binding pocket. We suggest that suramin analogs exert anti-adenoviral activities through a similar mechanism. However, structural studies will be required to localize the exact binding site(s) of suramin and its analogs in the 37FK. This will help to establish the structure–activity relationship and to design more-potent, small, sulfate-based inhibitors.

There are more than 80 HAdV types identified to date and these are classified into seven species (*A* to *G*) [41]. Several studies have shown that some antiviral compounds display anti-adenoviral activities against most/many HAdV types, whereas other compounds exert HAdV species-specific activity [10,11]. Thus, we wanted to investigate if the antiviral activity of suramin is HAdV species-specific. To test this, we pre-incubated HAdV-C5 (belonging to species C HAdV, and used here as a reference virus) with suramin, and analyzed virus infection of HCE cells. Interestingly, suramin inhibited (by ~85%) HAdV-D37 but not HAdV-C5 infection of HCE cells (Figure 4), indicating a HAdV species-specific antiviral activity of suramin. Heparin (used as a control glycan) neither inhibited HAdV-C5 nor HAdV-D37 infection of cells, as reported recently [26].

We speculate that the lack of antiviral activity of suramin against HAdV-C5 is because HAdV-C5 binds to a proteinaceous receptor (i.e., coxsackie and adenovirus receptor (CAR)) [20]. Unlike 37FK, 5FK mediates interaction with CAR receptors through loops that are located on the side of the FK, and this interaction is independent of charge [42]. We also assume that the overall positive charge of the viral FK contributes to the specific antiviral effect of suramin against species D HAdV-D37. 5FK has a pI of 5.67 (Table 1), which results in an overall slightly negative charge on the FK and may not allow charge-dependent interaction with suramin. Amino acid sequence analysis revealed that most EKC-causing HAdVs contain overall positively charged FKs (Table 1), thus we suggest that suramin or suramin-like molecules may display antiviral activity against all EKC-causing species D HAdVs.

Glycan-based molecules can function as soluble decoys and have promising therapeutic potential against multiple glycan-binding viruses [18,19]. Recently, Kwon et al. showed that nanodiscs carrying SA-containing glycans act as soluble decoys for influenza viruses (IV) [43]. These nanodiscs trap invading IVs and consequently inhibit virus infection both in vitro and in vivo. Similarly, Cagno et al. demonstrated the broad-spectrum antiviral activity of nanoparticles that display HS-proteoglycans [44]. Nanodiscs and/or nanoparticles carrying suramin or suramin-like molecules may also exhibit antiviral activity against EKC-causing HAdVs. The inhibitory efficiencies of glycan-based molecules often depend on the size and density of the charge present on these molecules. In line with this, we have recently shown that heparin polysaccharides inhibit HAdV-D37 binding to HCE cells more efficiently than its corresponding oligosaccharides [26]. Thus, we suggested that heparin and/or heparin-like molecules can act as soluble decoys and can be used as antivirals against HAdV-D37. Ideally, antiviral molecules would also need to penetrate tight junctions to reach the intercellular space and prevent the spreading of viruses from cell-to-cell. The polymeric nature and the large size of some glycan-based molecules may prevent these molecules from reaching the intercellular space. Thus, usage of small, sulfate-based compounds such as suramin will be advantageous as antivirals, especially against EKC-causing HAdVs. Such antivirals would not require systemic administration and could be administered topically (e.g., as eye drops). This will circumvent the poor pharmacokinetic profiles associated with glycan-based drugs that include fast serum clearance and poor cellular uptake.

In summary, we demonstrated for the first time that virus binding to cell surface decoy receptors constitutes a novel target for the development of attachment inhibitors. We suggest that small, sulfate, and non-glycan based molecules should be considered as antivirals against EKC-causing HAdVs as well as other glycan-binding ocular viruses. Such inhibitors offer various advantages. First, the anti-adenoviral mode of action of these inhibitors is on the extracellular level, which minimizes the risk of side effects. Most antiviral drugs target intracellular factors and often lead to side effects. Second, they block the very first step of the viral infection cycle, which limits the virus replication and progeny virus production. Third, they can enter the intercellular space of multilayer epithelial tissues (e.g., in the cornea) and thereby limit cell-to-cell virus transmission.

## Figures and Tables

**Figure 1 viruses-11-00242-f001:**
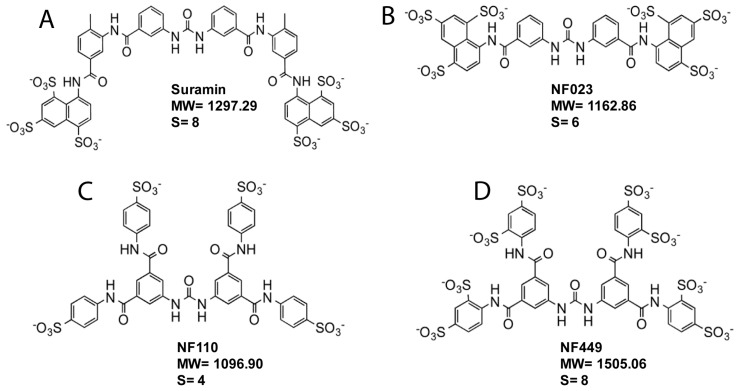
Structure of suramin and suramin analogs. (**A**) Suramin. (**B**) NF023. (**C**) NF110. (**D**) NF449. S = number of sulfate groups. MW = molecule weight.

**Figure 2 viruses-11-00242-f002:**
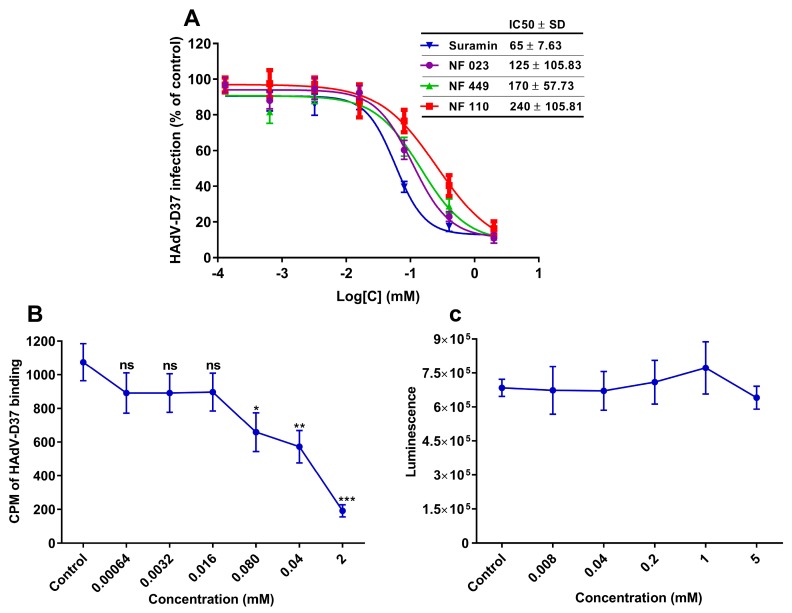
Dose-dependent inhibition of HAdV-D37 infection of, and binding to, human corneal epithelial (HCE) cells by GAG-mimetics. (**A**) HAdV-D37, pre-incubated with increasing concentrations of suramin and suramin analogs, infection of HCE cells. Virus infection of cells is presented as % infection of untreated viruses. (**B**) ^35^S-labeled HAdV-D37, pre-incubated with increasing concentrations of suramin, binding to HCE cells. Virus binding to cells is presented as count per minute (CPM). (**C**) Analysis of the cytotoxic effect of suramin on HCE cells. Cell viability was determined by using ATP-based assay and the result is presented as luminescence. Error bars represent mean ± SEM. ns = not significant. * *P* < 0.05, ** *P* < 0.01 and *** *P* < 0.001 relative to control.

**Figure 3 viruses-11-00242-f003:**
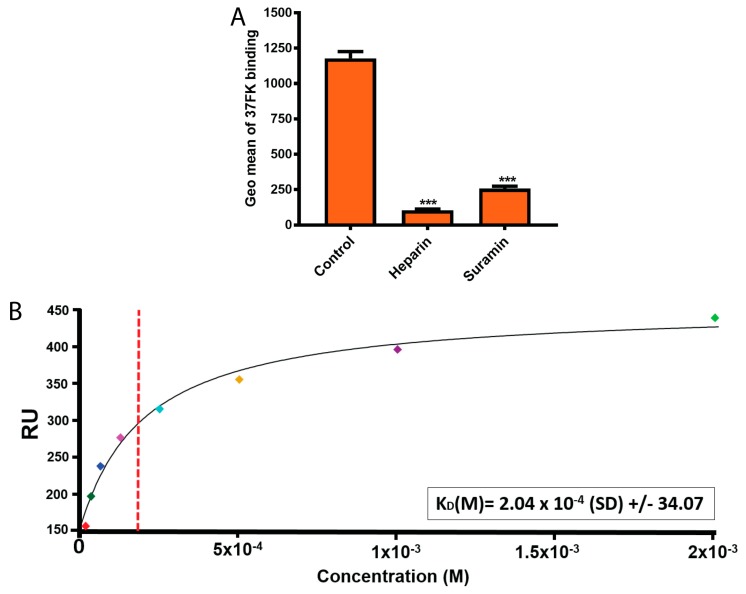
Suramin directly interacts with 37FK. (**A**) Binding of His-tagged 37FK, pre-incubated with heparin and suramin, to HCE cells. FK binding to cells was analyzed on the flow cytometer and the data is presented as the geometric mean of fluorescence (Geo mean). (**B**) SPR analysis of GST-tagged 37FK interaction to suramin. The red dashed vertical line represents the estimate of KD at half-maximal binding of the 37FK and suramin. The small squares in different colors represent the concentration points. Error bars represent mean ± SEM. *** *P* < 0.001 relative to control.

**Figure 4 viruses-11-00242-f004:**
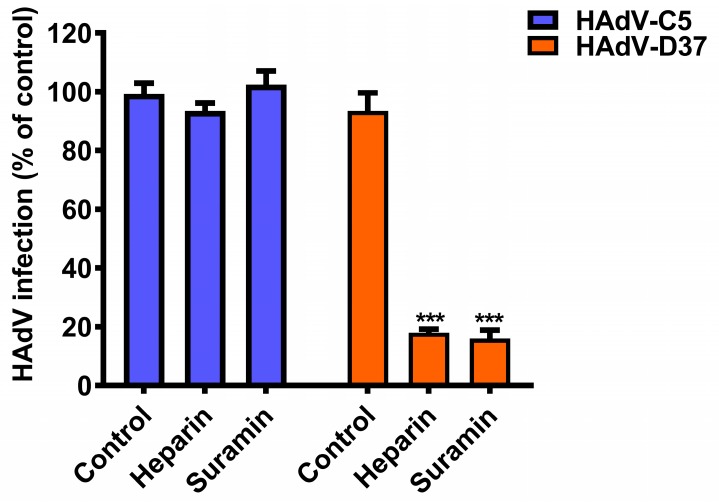
Antiviral activity of suramin is HAdV-type specific. HAdV-C5 and -D37 infection of human corneal (HCE) cells. Viruses were pre-incubated with heparin (0.4 mM) and suramin (1 mM). Virus infection of cells was quantified by immunofluorescence and the data is presented as % infection of untreated viruses. Error bars represent mean ± SEM. ns = not significant. *** *P* < 0.001 relative to control.

**Table 1 viruses-11-00242-t001:** Theoretical isoelectric points (pIs) of the FK domain of EKC-causing HAdVs.

HAdVs	Theoretical pI
HAdV-C5	5.67
HAdV-D8	9.04
HAdV-D37	9.14
HAdV-D53	9.04
HAdV-D54	8.88
HAdV-D56	7.80
HAdV-D64*	9.14

* Previously known as HAdV-19a.

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
