# Peer review of "Decoy Receptor Interactions as Novel Drug Targets against EKC-Causing Human Adenovirus"

_viruses, 2019, doi:10.3390/v11030242_

Reviewer 1 Report

The authors have provide enough experimental evidence that supports the capability of suramin to prevent or reduce HAd-37 infection. This study represents a potential antiviral drug compound for the treatment of Epidemic keratoconjunctivitis (EKC). This reviewer did not find any major concerns. 

Author Response

Comments and Suggestions for Authors:

The authors have provide enough experimental evidence that supports the capability of suramin to prevent or reduce HAd-37 infection. This study represents a potential antiviral drug compound for the treatment of Epidemic keratoconjunctivitis (EKC). This reviewer did not find any major concerns. 

Author's response:

- Since there were no additional comments from this reviewer, there is not much to respond to. We are happy that the reviewer like the manuscript.

Reviewer 2 Report

This study describes the use of suramin and analogues for inhibiting the binding of epidemic keratoconjunctivitis-causing HAdV-Ds to human corneal epithelial cells. This could provide an AdV species-specific inhibition strategy that may be explored in EKC patients.

The studies are carefully presented and support the conclusions. Nevertheless this reviewer has one suggestion for an additional experiment.

Would it be possible to study if suramin also inhibits cell-to-cell spread in a monolayer culture? I think there is no clarity yet whether such cell-to-cell spread is dependent on sialic acids as is the infection of cells by free Ads. Would it be possible to perform an experiment in which cells are infected by HAdV-37, and the suramin is added some 8 hrs later to monitor the effect on cell to cell spread, e.g. by monitoring plaque size? If this is possible please consider including such data.

Author Response

Comments and Suggestions for Authors:

This study describes the use of suramin and analogues for inhibiting the binding of epidemic keratoconjunctivitis-causing HAdV-Ds to human corneal epithelial cells. This could provide an AdV species-specific inhibition strategy that may be explored in EKC patients.

The studies are carefully presented and support the conclusions. Nevertheless this reviewer has one suggestion for an additional experiment.

Would it be possible to study if suramin also inhibits cell-to-cell spread in a monolayer culture? I think there is no clarity yet whether such cell-to-cell spread is dependent on sialic acids as is the infection of cells by free Ads. Would it be possible to perform an experiment in which cells are infected by HAdV-37, and the suramin is added some 8 hrs later to monitor the effect on cell to cell spread, e.g. by monitoring plaque size? If this is possible please consider including such data.

Author's response: We are happy that this reviewer suggest this experiment. This is something that we have considered, and we have also considered to perform a similar experiment in a multilayer model that we have recently developed (Storm et al J Virol 2017). However, we feel that such experiments and such a story is beyond the scope of the current manuscript, and will rather form the basis of another, future manuscript. The aim of the current manuscript was rather to show antiviral effect of decoy receptor analogues, and to identify the viral target molecule for these analogues.